# Beyond Isolated Words: Diffusion Brush for Handwritten Text-Line Generation

## Abstract

Existing handwritten text generation methods typically focus on isolated words. However, realistic handwritten texts require attention not only to individual words but also to the relationships between them, such as vertical alignment and horizontal spacing. Therefore, generating entire text line is a more promising task. However, this task poses significant challenges, such as accurately capturing complex style patterns including both intra-word and inter-word patterns, and maintaining content structure across numerous characters. To address these challenges, inspired by human writing priors, we focus on both the vertical style (*e.g.*, word alignment) and horizontal style (*e.g.*, word spacing and letter connections) of individual writing samples. Additionally, we decompose text-line content preservation across numerous characters into global context supervision between characters and local supervision of individual character structures. In light of this, we propose DiffBrush, a new diffusion model for text-line generation. DiffBrush employs two complementary proxy objectives to handle vertical and horizontal writing styles, and introduces two-level discriminators to provide content supervision at both the text-line and word levels. Extensive experiments show that DiffBrush excels in generating high-quality text-lines, particularly in style reproduction and content preservation. Our source code will be made publicly available.

## 1 Introduction

Handwritten text, as a remarkable symbol of human civilization, has recorded the history of human society from ancient times to the present. Even today, handwriting is considered a distinctly human skill. In the digital age, handwritten text generation merges the personalization of traditional writing with the efficiency of automation, garnering considerable interest. This task aims to automatically synthesize realistic handwritten text images that visually convey the user's unique writing style while ensuring the content readability. This can assist individuals facing writing difficulties, accelerate handwritten font design, and generate sufficient data to train more robust text recognizers.

Current dominant methods for this task generate handwriting images at word levels. For instance, some GAN-based methods (Bhunia et al., 2021; Gan et al., 2022; Pippi et al., 2023a) and diffusion-based method (Dai et al., 2024) utilize reference images provided by writers as style inputs and condition on character-wise labels or images for content inputs, achieving the synthesis of handwritten words with controllable styles and specified contents. However, as shown in Figure 1, we observe that handwritten text generation at word levels does not truly reflect the human writing process: 1) Humans generally maintain vertical alignment between words, while synthesized words often have arbitrary positions in the vertical aspect. 2) Different writers exhibit distinct horizontal word spacing, but this information is lost in the generated words. To address these issues, an intuitive solution is to generate entire text-lines rather than isolated words, known as handwritten text-line generation.

Our goal is to achieve high-quality handwritten text-line generation with desired styles and contents. The generation on text-line level, nevertheless, is very challenging due to several reasons: 1) It is non-trivial to accurately capture writing styles from text-lines with multiple words, as it involves not only intra-word style patterns like letter connections and slant but also inter-word spacing and vertical alignment. 2) Ensuring the readability of generated text-lines with numerous characters is difficult; for example, in the widely used IAM dataset (Marti & Bunke, 2002), a text-line averages 42 characters, roughly 6 times the length of a typical word.

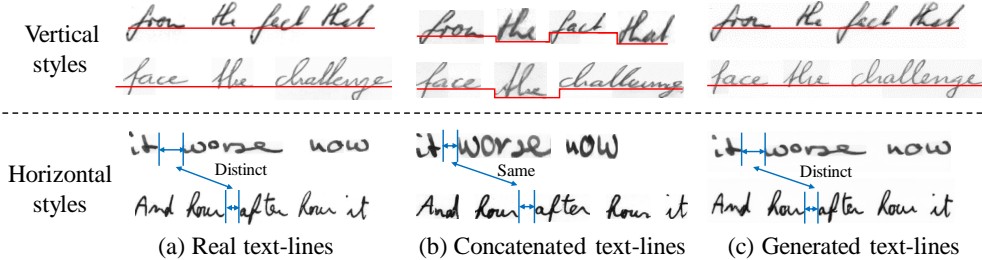

Vertical styles

Horizontal styles

(a) Real text-lines     (b) Concatenated text-lines     (c) Generated text-lines

Figure 1: Comparison of different handwritten text-lines: (a) Written by real writers. (b) Assembled with generated isolated words from One-DM (Dai et al., 2024), where fixed word spacing is applied due to the lack of spacing information in generated words. (c) Directly generated by our DiffBrush. Red lines indicate the baseline (i.e., the reference line at the bottom of the characters), while blue lines highlight word spacing. We observe that real text lines exhibit both vertical styles (*e.g.*, vertical alignment of words) and horizontal styles (*e.g.*, word spacing and letter connections). However, the isolated words do not accurately reproduce certain style patterns, such as vertical alignment and spacing between words. In contrast, our DiffBrush effectively captures these style patterns.

Previously, several GAN-based methods targeting text-line generation have been developed. TS-GAN (Davis et al., 2020) enhances the style vector by concatenating global and character-wise style features. However, the character-wise feature relies heavily on an independent character recognizer, making it difficult to capture all character styles accurately when the text-line contains many character categories. Moreover, TS-GAN naively uses text recognizers with CTC loss (Graves et al., 2006) for content supervision, which inadvertently hinders their style mimicry abilities. More specifically, to minimize the CTC loss, the model is pushed to generate easily recognizable samples with simple styles (*e.g.,* regular fonts with standard strokes). CSA-GAN (Kang et al., 2021) achieves handwritten text-line generation by introducing new data preprocessing and training strategies into GANWriting (Kang et al., 2020), which focuses on handwritten word generation. Nonetheless, CSA-GAN exhibits poor style learning capability since it directly uses a vanilla CNN as its style encoder.

Different from them, our solution is inspired by human writing priors and is built around two key principles: 1) People naturally pay attention to both vertical and horizontal styles of handwriting, as illustrated in Figure 1. The *vertical style* refers to the alignment of words along the vertical axis, while the *horizontal style* includes spacing between words, joins between letters, *etc*. 2) To ensure the content accuracy of handwritten text, *at a global level*, people maintain the correct character order within a text line, preserving global contextual relationships between characters. *At a local level*, they ensure the structural correctness of each individual word.

Guided by the above human writing priors, we propose DiffBrush, a diffusion model for handwritten text-line generation, featuring a dual-head style module and two-level content discriminators. Specifically, we employ the proxy loss (Movshovitz-Attias et al., 2017; Kim et al., 2020) to guide each head to focus on horizontal and vertical styles, respectively. For the vertical style, we randomly sample style references by column, preserving vertical alignment while disrupting word spacing and cursive connections, as shown in (a) of Figure 2. We then pull together column-wise sampling results from the same writer and push apart those from different writers, allowing the encoder to capture discriminative vertical style features. Similarly, for the horizontal style, we sample by row, retaining word spacing and cursive connections, as illustrated in (b) of Figure 2, and aggregate row-wise sampling results from the same writer to encourage the encoder to learn horizontal style patterns.

Furthermore, the proposed two-level content discriminators supervise textual content at both the line and word levels (cf. Figure 4). The line-level content discriminator segments the text-line image into non-overlapping parts, which are fed into a 3D CNN (Tran et al., 2015) to extract global contextual relationships. By assessing the realism of these relationships, the diffusion generator is encouraged to produce text-lines with correct character order. The word-level discriminator uses an attention mechanism to isolate individual words from the whole text-line and verify their content authenticity, guiding the generator to focus on text content at the local level. Our findings show that the two-level content discriminators improve content accuracy without reducing style imitation performance.

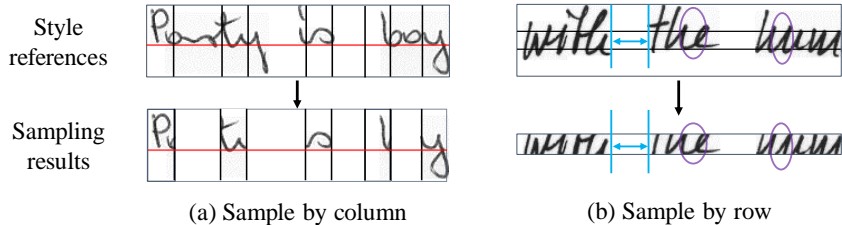

|  |  |
| :---: | :---: |
| (a) Sample by column | (b) Sample by row |

Figure 2: Two sampling strategies for style references. Red lines indicate vertical alignment between characters, while purple circles and blue lines highlight cursive connections between characters and spacing between words, respectively. In (a), a column-wise random sampling of style references preserves vertical alignment while disrupting word spacing and cursive connections. In contrast, (b) a row-wise sampling retains both word spacing and cursive joins.

We summarize our contributions in three key areas: 1) We propose DiffBrush, a new diffusion model targeting handwritten text-line generation. To the best of our knowledge, we are the first to explore how to design a diffusion model for handwritten text-line generation. 2) Inspired by two human writing priors, we propose a dual-head style module, which captures both vertical and horizontal writing styles, and two-level content discriminators that supervise textual content at both line and word levels while preserving style imitation performance. 3) Extensive experiments on two popular handwritten datasets demonstrate our DiffBrush significantly outperforms state-of-the-art methods.

## 2 RELATED WORK

**Handwritten Text Generation.** Handwritten text generation methods are generally divided into online and offline: the former synthesizes dynamic stroke sequences, while the latter generates static text images. Benefiting from the rapid advancement of deep learning, Recurrent Neural Networks (Kotani et al., 2020; Zhao et al., 2020; Tolosana et al., 2021), Transformer decoders (Dai et al., 2023), and diffusion models (Luhman & Luhman, 2020; Ren et al., 2023) have been widely used for synthesizing online handwritten text. However, online methods cannot synthesize stroke width, ink color, or paper background like offline methods.

The advent of Generative Adversarial Networks has accelerated the development of offline handwritten text generation. Early works (Alonso et al., 2019; Fogel et al., 2020) use character labels as content inputs and random noise as style inputs to synthesize handwritten words with controllable content and random styles. To enhance style control, SLOGAN (Luo et al., 2022) conditions style inputs on fixed writer IDs but fails to mimic unseen styles. Unlike them, GANwriting (Kang et al., 2020) and HWT (Bhunia et al., 2021) employ CNN or transformer encoder to extract style features from style references and are thus capable of imitating any styles. Further, VATr (Pippi et al., 2023a) utilizes symbol images as content representations, enabling character generation beyond the training charset. In contrast to the above word-focused methods, TS-GAN (Davis et al., 2020) and CSA-GAN (Kang et al., 2021) are developed to synthesize handwritten text-lines. However, they struggle to produce satisfactory results due to design drawbacks in style learning and content supervision.

**Image Diffusion.** Diffusion models such as Denoising Diffusion Probabilistic Model (DDPM) (Ho et al., 2020) and Latent Diffusion Model (LDM) (Rombach et al., 2022) have shown great success in image generation. For example, guided diffusion (Dhariwal & Nichol, 2021) and classifier-free diffusion (Ho & Salimans, 2022) condition the image synthesis on class labels. Some text-to-image diffusion methods like Stable-diffusion (Rombach et al., 2022) and DALL-E3 (Betker et al., 2023) further employ CLIP (Radford et al., 2021) to convert text descriptions into comprehensive representations, thereby producing impressive results. Very recently, some methods (Wang et al., 2023; Xu et al., 2024) combine adversarial learning with diffusion using a discriminator to enhance generation quality. Unlike these GAN-diffusion approaches that simply distinguish between real and generated images, our two-level content discriminators are specifically designed to provide content supervision at both the line and word levels.

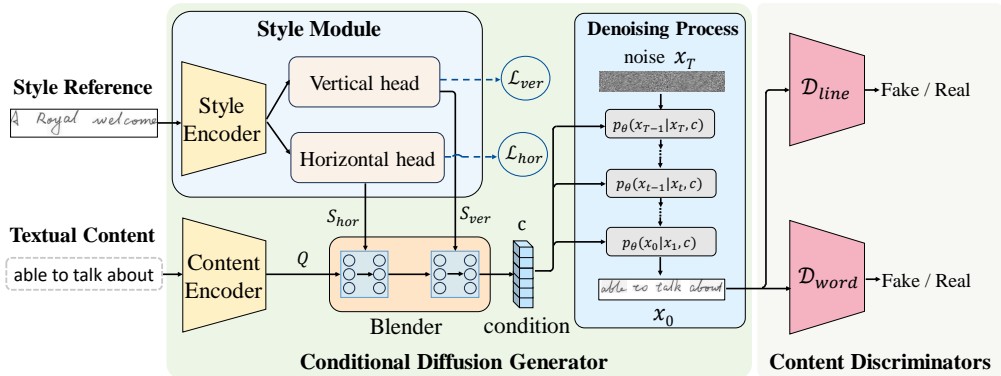

Figure 3: Overview of the proposed method. Our DiffBrush consists of a conditional diffusion generator and two-level content discriminators. Within the generator, vertical and horizontal style features captured by style module, together with content information extracted by content encoder, are fed into blender to obtain condition vector $c$. This condition is then used to guide the denoising process to generate the desired image $x_0$. We utilize the VerticalPA $\mathcal{L}_{ver}$ and HorizontalPA $\mathcal{L}_{hor}$ to force each head to extract its corresponding styles. The content discriminators provide content feedback at both the line and word levels to the generator for ensuring the readability of $x_0$.

The rapid development of diffusion models offers new potential for handwritten text generation task. However, some early attempts (Zhu et al., 2023; Nikolaidou et al., 2023) that condition denoising process on the fixed writer labels fail to mimic unseen handwriting styles. To address this, One-DM (Dai et al., 2024) extracts style information from both the writers' reference images and the high-frequency components of these images, then merges this information with the textual content to guide the denoising process, thereby enabling high-quality handwritten word generation. To our knowledge, developing a diffusion model for handwritten text-line generation remains unexplored.

## 3 METHOD

**Problem formulation.** We consider handwritten text-line image generation that is conditioned on both content and style. Given a text string $\mathcal{A}$ and a style reference $s_i$ randomly sampled from an exemplar writer $w_i \in \mathcal{W}$, we aim to synthesize a handwritten text-line image $x$ that captures the unique calligraphic style of $w_i$ while accurately preserving the content of $\mathcal{A}$. Here, $\mathcal{A} = \{a_i\}_{i=1}^L$ represents a sequence of length $L$, where each $a_i$ is a Unicode character, including lowercase and uppercase letters, digits, punctuation, *etc*. The key challenges lie in accurately capturing individual handwriting styles, including both intra-word and inter-word patterns from the style reference, while ensuring the readability of text lines that typically contain numerous characters.

To address this task, drawing inspiration from human writing principles related to style and content, we propose to capture both *vertical* and *horizontal* styles (cf. Figure 1) from individual handwritten examples while focusing on textual content at both the line and word levels. To achieve this, we introduce a novel DiffBrush method that innovates a dual-head style module with its distinct proxy losses, and the two-level content discriminators. Our DiffBrush can effectively imitate style patterns from the style reference, ensuring that the generated text-lines remain human-readable.

### 3.1 OVERALL SCHEME

The proposed DiffBrush (cf. Figure 3) comprises a conditional diffusion generator and two-level content discriminators. Within the conditional diffusion generator, the dual-head style module aims to emulate the vertical and horizontal styles of exemplar writers. To achieve this, we first employ a CNN-Transformer style encoder to extract rich calligraphic attributes from the provided style reference $s_i$. The vertical and horizontal heads then capture the respective styles from the extracted patterns. To guide this process, we introduce two proxy losses, VerticalPA $\mathcal{L}_{ver}$ and HorizontalPA $\mathcal{L}_{hor}$, which enforce each head to focus on its corresponding style. Specifically, $\mathcal{L}_{ver}$ brings closer the column-wise sampling results (cf. Figure 2) from the same writer, while $\mathcal{L}_{hor}$ aggregates the row-

wise sampling results (cf. Figure 2) belonging to the same writer. Through $\mathcal{L}_{ver}$ and $\mathcal{L}_{hor}$, the two heads obtain discriminative vertical and horizontal style features, *i.e.*, $S_{ver}$ and $S_{hor}$, respectively.

Considering the textual content, following VATr (Pippi et al., 2023a) and One-DM (Dai et al., 2024), we render the text string $\mathcal{A}$ into Unifont images. The strength of Unifont is its ability to represent all Unicode characters, allowing our method to accept any user-provided string input. We then input the rendered images into a content encoder with a CNN-Transformer architecture to obtain an informative content feature $Q = \{q_i\}_{i=1}^{L} \in \mathbb{R}^{L \times c}$ with contextual relationships, where $c$ is the channel dimension. After obtaining $Q$ and the two style representations $S_{ver}$ and $S_{hor}$, motivated by One-DM (Dai et al., 2024), we feed them into a blender with multi-head attention layers (Vaswani et al., 2017) for seamless fusion to obtain the conditional information $c \in \mathbb{R}^{L \times c}$. Specifically, we use $S_{ver}$ and $S_{hor}$ as key/value vectors and $Q$ as the query vector, successively attending $S_{ver}$ and $S_{hor}$ to aggregate the style information adaptively.

Guided by the fused condition $c$, the denoising network $p_\theta$ initiates the denoising process, where $\theta$ denotes the learnable parameters. Built on a U-Net architecture (Ronneberger et al., 2015), $p_\theta$ progressively synthesizes the desired handwritten text-line image $x_0$, starting from pure Gaussian noise $x_T \sim \mathcal{N}(0, \mathcal{I})$. The denoising process is supervised by a diffusion loss $\mathcal{L}_{diff}$ that minimizes the mean square error (MSE) between the generated $x_0$ and real $x_{real}$. However, relying solely on $\mathcal{L}_{diff}$ is insufficient to ensure the readability of the generated content. Therefore, two-level discriminators (i.e., $\mathcal{D}_{line}$ and $\mathcal{D}_{word}$) are introduced to provide content feedback.

Specifically, the conditional diffusion generator $G$ and the two-level discriminators $D$ engage in an adversarial learning process: $\mathcal{G}$ seeks to synthesize realistic images that $\mathcal{D}$ cannot distinguish from real ones based on content, while $\mathcal{D}$ assess the content at both the line and word levels. The readability of the generated images improves through two adversarial losses, $\mathcal{L}_{line}$ and $\mathcal{L}_{word}$, which further enhances generation quality in terms of content accuracy.

In summary, the overall training objectives for the conditional diffusion generator and the two-level discriminators are defined as follows:

$$\mathcal{L}_{\mathcal{G}} = \mathcal{L}_{ver} + \mathcal{L}_{hor} + \mathcal{L}_{diff} + \lambda(\mathcal{L}_{line} + \mathcal{L}_{word}), \tag{1}$$

$$\mathcal{L}_{\mathcal{D}} = -(\mathcal{L}_{line} + \mathcal{L}_{word}), \tag{2}$$

where $\lambda$ serves as a trade-off factor. We alternately optimize $\mathcal{G}$ and $\mathcal{D}$, and experimentally set $\lambda$ to 0.05 in the training phase.

## 3.2 DUAL-HEAD STYLE MODULE

To capture complex style patterns within text-lines (cf. Figure 1), such as vertical alignment between words and horizontal word spacing, we propose a dual-head style module to extract both vertical and horizontal styles from individual reference samples. As illustrated in Figure 3, the style samples are first fed into a style encoder, which combines a CNN and a transformer encoder, to obtain an initial style feature sequence $S \in \mathbb{R}^{d \times c}$, where $d$ is the sequence length. Subsequently, we employ two separate heads, termed vertical head and horizontal head, each containing a standard self-attention layer, to extract vertical style $S_{ver} \in \mathbb{R}^{d \times c}$ guided by $\mathcal{L}_{ver}$ and horizontal style $S_{hor} \in \mathbb{R}^{d \times c}$ guided by $\mathcal{L}_{hor}$ from $S$, respectively. The details of $\mathcal{L}_{ver}$ and $\mathcal{L}_{hor}$ are detailed below.

**Vertical Style Learning.** The goal of the proposed $\mathcal{L}_{ver}$ is to guide the vertical head in extracting the discriminative vertical style $S_{ver}$. However, accurately learning the vertical style is challenging because the samples inherently contain both vertical and horizontal style patterns. To address this issue, we propose to draw together column-wise sampling results of reference samples from the same writer, thereby enforcing the vertical head to learn $S_{ver}$. The intuition is that the column-wise sampling process maintains vertical alignment between characters while disrupting horizontal style patterns such as word spacing and cursive connections (cf. Figure 2).

To implement this, we divide the style image into several columns and then randomly select a subset following a uniform distribution. More specifically, we perform the sampling process on $S_{ver}$ by first reshaping the sequential feature $S_{ver}$ back into spatial feature $\hat{S}_{ver} \in \mathbb{R}^{h \times w \times c}$, and then sampling columns of $\hat{S}_{ver}$ to obtain $s_{col} \in \mathbb{R}^{h \times n \times c}$, where $n = w \cdot \rho$ and $\rho$ is the sampling ratio. Next, $\mathcal{L}_{ver}$ assigns a proxy to each writer, treating each proxy as an anchor and associating it with all column-

wise sampling results. Proxy offers faster convergence and avoids the need for complex data pair construction. We formulate our $\mathcal{L}_{ver}$ as follows:

$$
\begin{aligned}
\mathcal{L}_{ver} = &\frac{1}{|P_{col}^+|} \sum_{p_{col} \in P_{col}^+} \log\left(1 + \sum_{s_{col} \in S_{col}^+} e^{-\alpha(sim(f(s_{col}),p_{col})-\delta)}\right) \\
&+ \frac{1}{|P_{col}|} \sum_{p_{col} \in P_{col}} \log\left(1 + \sum_{s_{col} \in S_{col}^-} e^{\alpha(sim(f(s_{col}),p_{col})+\delta)}\right).
\end{aligned}
\tag{3}
$$

In detail, $S_{col} = \{s_{col}^i\}_{i=1}^N$ represents a mini-batch of length $N$. $P_{col}$ denotes the set of proxies corresponding to all writers, and $P_{col}^+$ refers to the set of writers present in the current batch. For each proxy $p_{col} \in \mathbb{R}^c$, $S_{col}$ is divided into a positive set $S_{col}^+$, consisting of $s_{col}$ from the same writer as $p_{col}$, and a negative set $S_{col}^- = S_{col} - S_{col}^+$. $f(\cdot)$ denotes the mean pooling operation and $sim(\cdot, \cdot)$ represents the cosine similarity between two vectors, $\delta > 0$ is a margin and $\alpha$ is a scaling factor.

**Horizontal Style Learning.** Unlike vertical style learning, $\mathcal{L}_{hor}$ aims to encourage the horizontal head for extracting the discriminative horizontal style $S_{hor}$. To achieve this, we focus on narrowing the gap between row-wise sampling results from the same writer. The row-wise sampling process preserves horizontal style patterns such as word spacing and cursive joins (cf. Figure 2).

We achieve this by dividing the style image into several rows and randomly selecting a subset based on a uniform distribution. Specifically, we reshape the sequential feature $S_{hor}$ back into a spatial feature $\hat{S}_{hor} \in \mathbb{R}^{h \times w \times c}$ and then sample rows to obtain $s_{row} \in \mathbb{R}^{m \times w \times c}$, where $m = h \cdot \rho$. Similar to vertical style learning, we assign a proxy $p_{row}$ to each writer and link it with all row-wise sampling results $S_{row}$ in a mini-batch. The HorizontalPA $\mathcal{L}_{hor}$ is formulated as:

$$
\begin{aligned}
\mathcal{L}_{hor} = &\frac{1}{|P_{row}^+|} \sum_{p_{row} \in P_{row}^+} \log\left(1 + \sum_{s_{row} \in S_{row}^+} e^{-\alpha(sim(f(s_{row}),p_{row})-\delta)}\right) \\
&+ \frac{1}{|P_{row}|} \sum_{p_{row} \in P_{row}} \log\left(1 + \sum_{s_{row} \in S_{row}^-} e^{\alpha(sim(f(s_{row}),p_{row})+\delta)}\right).
\end{aligned}
\tag{4}
$$

### 3.3 TWO-LEVEL CONTENT DISCRIMINATORS

Unlike existing methods (Davis et al., 2020; Gan et al., 2022; Pippi et al., 2023a; Dai et al., 2024) that simply employ recognizers with CTC loss to improve the content readability of generated images, we propose two-level discriminators focused on providing effective content feedback. The advantage of our discriminators is that they improve content accuracy without disrupting style learning, while CTC-based methods tend to hinder it. Our discriminators address two key challenges: (1) How to ensure that the discriminator focuses on textual content rather than style, and (2) Considering that a text line typically contains numerous characters, it is challenging to provide effective supervision to ensure the accurate generation of each character.

To address the challenge (1), inspired by pix2pix (Isola et al., 2017), we introduce textual content as a conditional input, feeding it into the discriminator alongside the generated image. This ensures that the discriminator focuses solely on content evaluation. For challenge (2), we break it down into two more simpler subtasks: assessing the correctness of the overall character order and verifying the correctness of the local text content. The proposed two-level discriminators consist of a text-level discriminator and a word-level discriminator Figure 4. We detail each component below.

**Line-level Content Discriminator.** Given the generated image $x_0$ and the content guidance $I_{line}$ without style information, the line-level discriminator $\mathcal{D}_{line}$ aims to determine whether the character order in $x_0$ aligns with that in $I_{line}$. Specifically, we concatenate $x_0$ and $I_{line}$ along the channel dimension, and then slice the concatenated result into $n$ non-overlapping segments $\{c_i\}_{i=1}^n$ from left to right. $\{c_i\}_{i=1}^n$ are processed by a 3D CNN to integrate context information, outputting $n$ patches. $\mathcal{D}_{line}$ then determines whether each patch is real or fake, providing fine-grained feedback on character order. The line-level discriminator loss $\mathcal{L}_{line}$ is formulated as:

$$
\mathcal{L}_{line} = log(\mathcal{D}_{line}(I_{line}, x_{real})) + log(1 - \mathcal{D}_{line}(I_{line}, x_0)). \tag{5}
$$

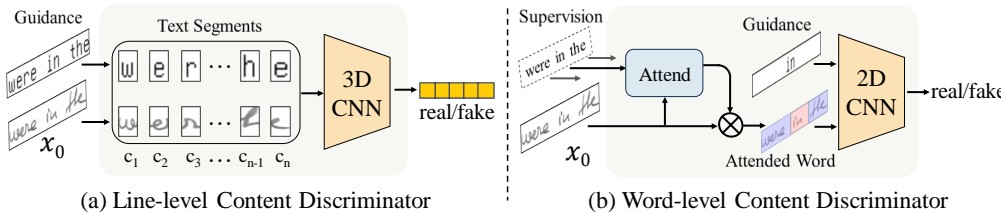

(a) Line-level Content Discriminator        (b) Word-level Content Discriminator

Figure 4: Illustration of the two-level content discriminators.

**Word-level Content Discriminator.** Compared to the line-level discriminator $\mathcal{D}_{line}$, the word-level discriminator $\mathcal{D}_{word}$ is designed to ensure that the text structure is correctly generated at the word level. However, accurately locating word positions within a whole text-line $x_0$ is non-trivial. Motivated by ASTER (Shi et al., 2018), we utilize an attention module with a CNN-LSTM architecture to obtain word positions. A CNN encoder first extracts spatial features $F_{map} \in \mathbb{R}^{h \times w \times c}$ from $x_0$, which are flattened into sequential features $H \in \mathbb{R}^{l \times c}$, where $l = h \times w$. The LSTM decoder then takes $x_0$ and a start-of-sequence (SOS) token as input, sequentially outputting attention maps for character positions until the end-of-sequence (EOS) token is reached.

Character-level attention maps are concatenated into word-level attention maps $A = \{a_t\}_{t=1}^{T}$, where $a_t \in \mathbb{R}^{h \times w}$, to extract attended words $\{x_{word}^t\}_{t=1}^{T}$, with $x_{word}^t = F_{map} \cdot a_t$ and $T$ being the number of words in the text-line. Finally, each $x_{word}$ and its corresponding content guidance $I_{word}$ are fed into $\mathcal{D}_{word}$ (cf. Figure 11 in Appendix). The generator is encouraged to refine the detailed structure of the generated images through the word-level discriminator loss $\mathcal{L}_{word}$:

$$\mathcal{L}_{word} = \sum_{i=1}^{T} log([\mathcal{D}_{word}(I_{word}, x_{real}^i)]_i) + \sum_{i=1}^{T} log(1 - [\mathcal{D}_{word}(I_{word}, x_{word}^i)]_i), \quad (6)$$

where $[\mathcal{D}_{word}(\cdot, \cdot)]_i$ represents the discrimination output for the $i$-th word within a full-line text.

## 4 EXPERIMENTS

### 4.1 EXPERIMENTAL SETTINGS

**Evaluation Dataset.** To evaluate our DiffBrush in generating handwritten text-line, we use the widely adopted handwriting dataset IAM (Marti & Bunke, 2002) and CVL (Kleber et al., 2013). IAM contains 13,353 English text-line images belonging to 657 unique writers. Following the protocol of CSA-GAN (Kang et al., 2021), we use text-lines from 496 writers for training and the remaining 161 writers for testing. CVL dataset consists of handwritten text-lines from 310 writers in both English and German. For our experiments, we use the English portion, consisting of 11,007 text-lines, and follow the standard CVL split, with 283 writers for training and 27 for testing. In all experiments, we resize the images to a height of 64 pixels while preserving their aspect ratio, as done in previous works(Davis et al., 2020; Kang et al., 2021; Dai et al., 2024). To manage varying widths, images with a width smaller than 1024 pixels are padded, whereas those exceeding 1024 pixels are resized to a fixed size of 64 × 1024. We also conduct user studies to quantify the subjective quality of the generated handwritten text-line images in Appendix A.2.

**Evaluation Metrics.** 1) We use the newly proposed Handwriting Distance (HWD) (Pippi et al., 2023b), specifically designed for handwriting style evaluation. HWD computes the Euclidean distance between features extracted by a VGG16 network pre-trained on a large corpus of handwritten text images. 2) We evaluate content accuracy using an OCR system, following CSA-GAN (Kang et al., 2021) and One-DM (Dai et al., 2024). 3) We use Fréchet Inception Distance (FID) (Heusel et al., 2017), Inception Score (IS) (Salimans et al., 2016), and Geometry Score (GS) (Khrulkov & Oseledets, 2018) to measure the visual quality of generated images.

**Implementation details.** In all experiments, we use a randomly selected text-line sample as the style reference. In our DiffBrush, both the style and content encoders are based on a Resent18, followed by 2 standard transformer encoder layers. The blender has 6 transformer decoder layers

for receiving style representations (3 for vertical and 3 for horizontal). Line-level discriminator uses three 3D convolution layers, and word-level discriminator has three 2D convolution layers.

During training, we drop the condition $c$ with the probability $0.1$, following classifier-free diffusion (Ho & Salimans, 2022). The model is trained for 800 epochs on eight RTX 4090 GPUs using the AdamW optimizer with a learning rate of $10^{-4}$. For the sampling ratio $\rho$, we perform a grid search over $\{0.25, 0.5, 0.75, 1.00\}$ and ultimately set $\rho$ to 0.25. During sampling, we adopt a classifier-free strategy with the guidance scale of 0.2. For sampling, we adopt a classifier-free strategy with a guidance scale of 0.2 and use DDIM (Song et al., 2021) with 50 steps to accelerate the process. More details are provided in Appendix A.1.

**Compared Methods.** We compare DiffBrush with state-of-the-art handwritten text-line generation methods, including TS-GAN (Kang et al., 2021), CSA-GAN (Kang et al., 2021), and advanced handwritten text generation approaches like VATr (Pippi et al., 2023a) and One-DM (Dai et al., 2024). For a fair comparison, we retrain VATr and One-DM on the text-line datasets using their official implementations, enabling them to directly synthesize text-line images.

## 4.2 MAIN RESULTS

| Datasets | Method | Shot | HWD ↓ | CER ↓ | WER ↓ | FID ↓ | IS ↑ | GS ↓ |
|---|---|---|---|---|---|---|---|---|
| IAM | TS-GAN | one | 2.11 | 44.20 | 87.13 | 16.76 | 1.76 | $2.87 \times 10^{-2}$ |
| | CSA-GAN | few | 2.25 | 42.27 | 84.14 | 13.52 | 1.74 | $1.62 \times 10^{-2}$ |
| | VATr | few | 1.87 | 28.80 | 71.77 | 12.51 | 1.69 | $1.45 \times 10^{-2}$ |
| | One-DM | one | 1.80 | 20.91 | 54.27 | 10.60 | 1.82 | $8.42 \times 10^{-3}$ |
| | Ours | one | **1.41** | **8.59** | **28.60** | **8.69** | **1.85** | $\mathbf{2.35 \times 10^{-3}}$ |
| CVL | CSA-GAN | few | 1.72 | 41.64 | 72.02 | 8.71 | 1.48 | $6.71 \times 10^{-2}$ |
| | VATr | few | 1.5 | 38.49 | 66.33 | 9.04 | 1.44 | $1.43 \times 10^{-1}$ |
| | One-DM | one | 1.47 | 32.42 | 63.35 | 11.95 | 1.46 | $1.29 \times 10^{-1}$ |
| | Ours | one | **1.06** | **20.92** | **36.38** | **7.57** | **1.70** | $\mathbf{2.96 \times 10^{-2}}$ |

Table 1: Comparisons with state-of-the-art methods on handwritten text-line generation in the IAM and CVL datasets. All methods are trained on the same training set and evaluated using the same protocols. The "Shot" column indicates the number of style references required for each method.

**Styled Handwritten Text-line Generation.** Firstly, we assess our DiffBrush for generating handwritten text-line images with desired style and specific content. To quantify style similarity, following CSA-GAN (Kang et al., 2021), we generate text-line images for each method using style information from test set and content input from a subset of WikiText-103 (Merity et al., 2016). We then calculate the HWD between the generated and real samples for each writer, and finally average the results. For content evaluation, we use the generated training sets from each method to train an OCR system (Retsinas et al., 2022) and report its recognition performance on the real test set, as done in CSA-GAN (Kang et al., 2021) and One-DM (Dai et al., 2024).

The quantitative results in Table 1 show that DiffBrush outperforms all state-of-the-art methods on both IAM and CVL datasets. Specifically, it improves HWD by $21.67\%$ ($1.80 \rightarrow 1.41$) on IAM and $27.89\%$ ($1.47 \rightarrow 1.06$) on CVL compared to the second-best method, highlighting its superior style imitation ability. Moreover, DiffBrush achieves significantly lower CER and WER on both IAM and CVL datasets, further demonstrating its advantage in content readability.

We further provide qualitative results to intuitively explain the benefit of our DiffBrush in Figure 5. TS-GAN struggles to accurately capture the style patterns of reference samples, such as ink color and stroke width. CSA-GAN produces samples that lack style consistency, including inconsistent character slant, ink blot, and stroke width. VATr has difficulty maintaining vertical alignment between words in the synthesized text lines. One-DM occasionally generates text lines with missing or incorrect characters. In contrast, our DiffBrush excels at generating precise character details while maintaining overall consistency. We provide additional qualitative comparisons in Figure 12 through Figure 14 of Appendix.

**Style-agnostic Handwritten Text-line Generation.** We also evaluate DiffBrush's ability to generate realistic handwritten text-line images, independent of style imitation. Following TS-GAN (Davis

| | Style samples | TS-GAN | CSA-GAN | VATr | One-DM | Ours |

Figure 5: Qualitative comparisons between our method and state-of-the-art approaches for handwritten text-line generation with specific textual content and desired style on the IAM dataset. We use the same guiding text, "Success is not the destination, it's the journey, every step forward is a step forward growth. Believe in yourself, and anything is possible." for all methods, instructing them to generate the text in different handwriting styles. The red circles highlight missing characters or structural errors, while the blue circles emphasize detailed style inconsistencies, such as character slanting and ligatures. Better zoom in 200%.

| Style sample | | HWD↓ | CER↓ | WER↓ |
|---|---|---|---|---|
| Base | | 1.82 | 39.86 | 77.75 |
| Base+$\xi_{style}$ | | 1.47 | 38.26 | 75.96 |
| Base+$\xi_{style}$+$\mathcal{D}_{line}$ | | 1.44 | 15.28 | 43.31 |
| Base+$\xi_{style}$+$\mathcal{D}_{line}$+$\mathcal{D}_{word}$ | | 1.43 | 8.59 | 28.60 |

Figure 6: Ablation study on IAM dataset. Effect of style module $\xi_{style}$, and the line-level and word-level content discriminators, *i.e.*, $\mathcal{D}_{line}$ and $\mathcal{D}_{word}$. In the middle, we showcase the generated samples of each component. The red boxes highlight failures of structure preservation.

et al., 2020), each method generates 25k random text-line images to calculate FID against 25k cropped samples from the training set, and 5k random samples for GS calculation, compared with 5k samples from the test set. Besides, we generate the entire test set using each method and evaluate the results using the IS metric. As shown in Table 1, DiffBrush achieves the highest performance across FID, IS, and GS metrics on both IAM and CVL datasets, further demonstrating its ability to generate superior-quality handwritten text-line images.

## 4.3 ANALYSIS

In this section, we conduct ablation studies to analyze our DiffBrush. More analyses are provided in Appendix, including application for downstream task (*i.e.*, enrich datasets to train more robust recognizer) and failure case analysis.

**Quantitative evaluation of style module and content discriminators.** We perform multiple ablation studies on the IAM dataset to validate the effect of different components. We provide the quantitative result in Figure 6. We find that: (1) The introduction of style module leads to a significant 19.23% improvement in HWD (1.82 → 1.47), underscoring its effectiveness in style learning.

Figure 7: The red lines highlight misalignment of words along the vertical axis, while the blue circles indicate failures in capturing ligature patterns.

Figure 8: Style interpolation results between different individual handwriting styles on IAM dataset.

(2) The sequential integration of the line-level and word-level discriminators leads to significant improvements in terms of CER and WER without reducing HWD. This demonstrates that our discriminators enhance content readability while preserving style imitation performance.

**Qualitative evaluation of style module and content discriminators.** we conduct visual ablation experiments to further analyze each module in our DiffBrush. As shown in Figure 6, we observe that the base version shows clear drawbacks in both style imitation and content readability. Adding the style module significantly improves style reproduction, such as ink color and stroke width, but content readability remains poor. Introducing the line-level discriminator enhances overall content readability, but character detail issues still remain. Finally, adding the word-level discriminator resolves missing and unnecessary character problems, further improving content accuracy.

**Discussions about two style representations.** We conduct ablation experiments on the dual-head style module to analyze the differences between the two styles. As shown in Figure 7, removing either the vertical or horizontal styles reduces generation quality in terms of HWD. Specifically, removing the $\mathcal{L}_{ver}$ weakens the model's ability to capture vertical alignment, making it difficult to align words at a consistent height. On the other hand, removing the $\mathcal{L}_{hor}$ impairs the model's ability to capture horizontal features, such as word spacing and character ligatures.

**Discussions about the learned style space.** To further explore the latent space learned by our style module, we conduct linear style interpolation experiments between different writers and display the generated handwritten text-line images in Figure 8. From these visual results, we find that the generated text-line images smoothly transition from one style to another, in terms of character slant, and stroke thickness, while strictly preserving their original textual content. These results further demonstrate that our method effectively generalizes to the handwriting style latent space, rather than merely memorizing style patterns from individual handwriting samples.

## 5 CONCLUSION

In this paper, we introduce DiffBrush, a novel diffusion model tailored for handwritten text-line generation. To the best of our knowledge, this is the first exploration of diffusion models for this task. Drawing inspiration from two human writing priors, we propose a dual-head style module that captures both vertical and horizontal writing styles, and two-level content discriminators that supervise textual content at both the line and word levels while preserving style imitation performance. Promising results on two widely-used handwritten datasets verify the effectiveness of our DiffBrush. In the future, we plan to extend DiffBrush to support multi-script handwritten text generation.

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

# A APPENDIX

## A.1 MORE IMPLEMENTATION DETAILS

In our conditional diffusion generator, each Transformer layer contains the multi-head attention with $c = 512$ dimensional states and 8 attention heads. We apply sinusoidal positional encoding (Vaswani et al., 2017) to input tokens before feeding them to the Transformer encoder layer. We pre-train the blender on handwritten text-line recognition task with cross-entropy loss and fix its parameter during the training of the whole DiffBrush. To conserve GPU memory and accelerate the training time, following Wordstylist (Nikolaidou et al., 2023) and One-DM (Dai et al., 2024), we streamline the U-Net by reducing the number of ResNet blocks and attention heads and takes the diffusion process into the latent space. Specifically, we adopt a powerful, pre-trained Variational Autoencoder (VAE) of Stable Diffusion (1.5) to convert the image into latent space. During the training phase, we freeze the parameters of VAE and we set $T = 1000$ steps, and forward process variances are set to constants increasing linearly from $\beta_1 = 10^{-4}$ to $\beta_T = 0.02$.

## A.2 USER STUDIES

**User preference study.** We invite human participants with postgraduate education backgrounds to evaluate the visual quality of synthesized handwritten text images, focusing on style imitation. The generated samples are from our method and other state-of-the-art approaches. In each round, we randomly select a writer from the IAM dataset and use their handwritten text-line sample as style guidance, along with identical text as content guidance, to direct all methods in generating candidate samples. Participants are presented with one text-line from the exemplar writer as a style reference and multiple candidates generated by different methods. They are asked to select the candidate that best matches the reference in style. This process is repeated 30 times, yielding 900 valid responses from 30 volunteers. As shown in Figure 9, our method receives the most user preferences, demonstrating its superior quality in style imitation.

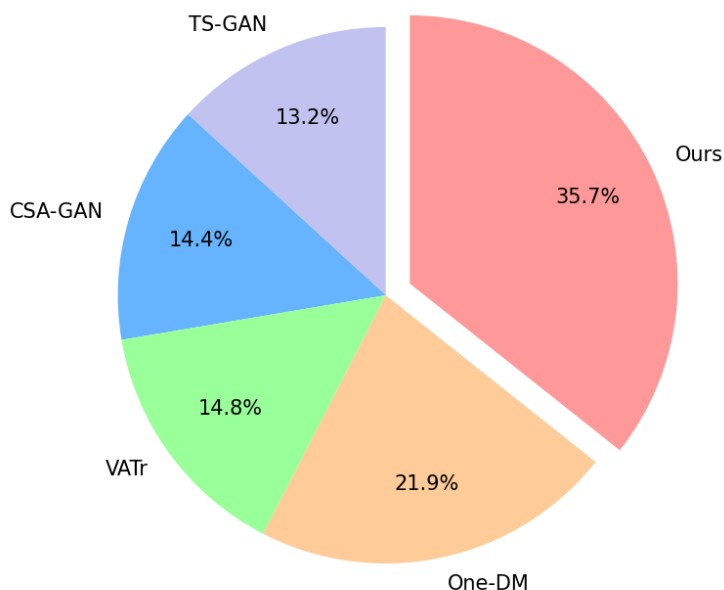

Figure 9: User preference study with a comparison to state-of-the-art methods on handwritten text-line generation.

**User plausibility study.** We conduct a user plausibility study to assess whether the text-line images generated by DiffBrush are indistinguishable from real handwriting samples. In this study, participants are first shown 30 examples of authentic handwritten text-line samples. They are then asked to classify each image they see as either real or synthetic, with the images being randomly selected from both genuine samples and those generated by our method. In total, 30 participants provide

Table 2: Confusion matrix(%) from the user plausibility study. The classification accuracy of 49.11% suggests that users struggle to differentiate between handwritten text-line images generated by our DiffBrush and real ones.

| Actual | Predicted | | Classification Accuracy |
|--------|------|------|------------------------|
|        | Real | Fake |                        |
| Real   | 27.22 | 22.78 | 49.11 |
| Fake   | 28.11 | 21.89 |       |

900 valid responses. The results, shown as a confusion matrix in Table 2, report a classification accuracy close to 50%, suggesting the task becomes equivalent to random guessing. This indicates that text-line images generated by our method are nearly indistinguishable from real samples.

## A.3 APPLICATION FOR RECOGNIZER PERFORMANCE IMPROVEMENT

A key application of handwritten text-line generation models is to enrich the training dataset, facilitating the training of more robust recognizers. To this end, we combine the IAM training set generated by various methods with the real training set to create a new mixed dataset. We then train an OCR system using this mixed dataset and report its performance on the real IAM test set. We present the quantitative results in the table. These results clearly show that the additional synthetic data contributes to improving the recognizer's performance. Among all methods, our approach achieves the greatest performance improvement, with an improvement rate of 20.07%.

| Training Data | CER ↓ | WER ↓ | Improvement Rate (%) ↑ |
|---------------|-------|-------|------------------------|
| Real | 5.78 | 21.76 | - |
| CSA-GAN + Real | 5.39 | 19.89 | 6.74 |
| VATr + Real | 5.08 | 19.31 | 12.11 |
| One-DM + Real | 4.99 | 18.51 | 13.67 |
| DiffBrush (Ours) + Real | **4.62** | **16.86** | **20.07** |

Table 3: Handwritten text-line recognition on different training data. Improvement rate refers to CER performance gain achieved by incorporating synthetic data into the training process compared to using only the real training set.

## A.4 ANALYSIS OF FAILURE CASES

We find that DiffBrush occasionally generates structurally incorrect characters when low-frequency characters from the training set are used as content conditions. This includes punctuation marks and Greek letters, as highlighted by the red circles in Figure 10. A simple yet effective solution is to employ a data oversampling strategy, increasing the frequency of these characters during training.

| | |
|---|---|
| Style sample | *Federal Government that the financial burden* |
| Text Content | " other " fish (+8 to -11) & "other" vegetables |
| VATr | *' other " fish [+8 to ;11)8 "other" vegetables* |
| One-DM | *"other " fish (78 to -11) & "other," vegetables* |
| Ours | *"other " fish (+8 to -11) 2 "other" vegetables* |
| Style sample | *bases in this country. An open letter* |
| Text Content | θ, μ stand for the angle, momentum parameter. |
| VATr | *2, y stand for the angle, momentum parameter.* |
| One-DM | *d. w stand for the angle, momentum parameter.* |
| Ours | *θ,y stand for the angle, momentum parameter.* |

Figure 10: Failure cases. The red circles highlight character structure errors.

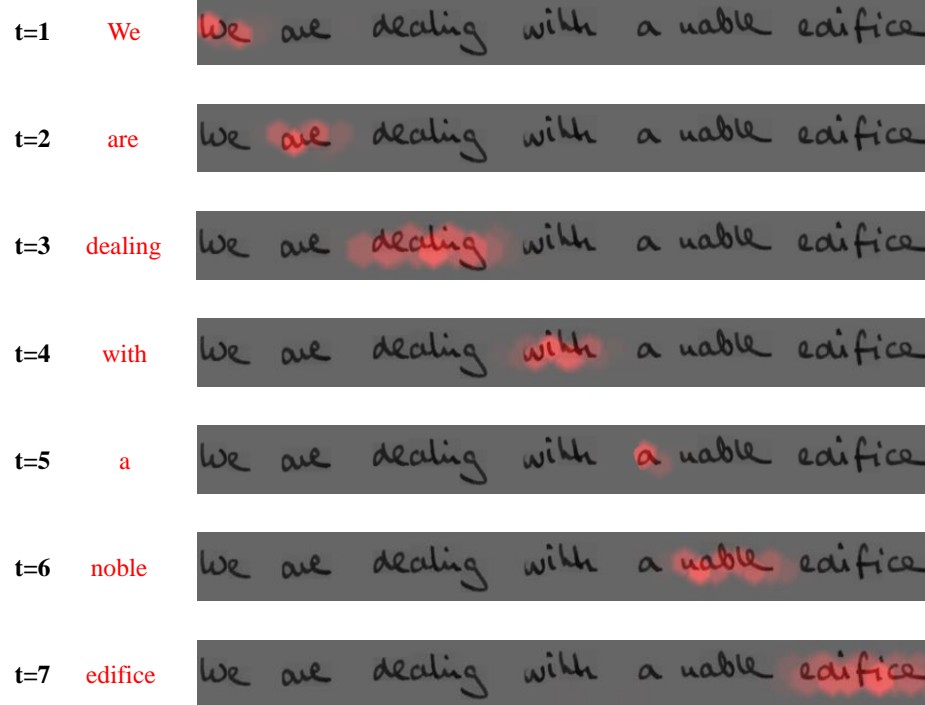

| | |
|---|---|
| **t=1** | We |
| **t=2** | are |
| **t=3** | dealing |
| **t=4** | with |
| **t=5** | a |
| **t=6** | noble |
| **t=7** | edifice |

Figure 11: Visualization of attention maps for each word in a text-line image.

Figure 12: Comparisons with the state-of-the-art methods for handwritten text-line generation. The green circles highlight inconsistencies in ink color compared to the given style reference.

| | |
|---|---|
| Style sample | *better to pay the fair price for a tool of good* |
| Text Content | Your limitation - it's only your imagination, push past it. |
| TS-GAN | *Your limitation - it's only your imagination, push past it.* |
| CSA-GAN | *Your limitation - it's only your imagination, push past it.* |
| VATr | *Your limitation i it's only your imagination, push past it.* |
| One-DM | *Your limitation - it's only your imagination, push past it.* |
| Ours | *Your limitation - it's only your imagination, push past it.* |
| Style sample | *well trounced by the critics wherever it* |
| Text Content | The future belongs to those who believe in their dreams. |
| TS-GAN | *The future belongs to those who believe in their dreams.* |
| CSA-GAN | *The future belongs to those who believe in their dreams.* |
| VATr | *the future belongs to those who believe in their dreams.* |
| One-DM | *The future belongs to thoswho believe in their dreams.* |
| Ours | *The future belongs to those who believe in their dreams.* |
| Style sample | *Today, for example, the Foreign Minister of Indo-* |
| Text Content | One day or day one-you decide, take action today. |
| TS-GAN | *One day or day one-you decide, take action today.* |
| CSA-GAN | *One day or day one-you decide, take action today.* |
| VATr | *One day or day one-you decide, take action today.* |
| One-DM | *One day or day one-you decide, take action today.* |
| Ours | *One day or day one-you decide, take action today.* |

Figure 13: Comparisons with the state-of-the-art methods for handwritten text-line generation. The blue circles highlight errors in ligatures, while the red circles emphasize incorrect content structure. The green circles highlight inconsistencies in ink color compared to the given style reference.

Figure 14: Comparisons with state-of-the-art methods for handwritten text-line generation. The red circles highlight incorrect content structure, while the green circles point out ink color inconsistencies relative to the style reference.

