# OpenReview forum: "Beyond Isolated Words: Diffusion Brush for Handwritten Text-Line Generation"
_ICLR.cc/2025/Conference — ICLR 2025 Conference Withdrawn Submission_

### Official Review · Reviewer_cLAK · 2024-10-28

**Soundness:** 3
**Presentation:** 3
**Contribution:** 2
**Rating:** 6
**Confidence:** 3

**Summary:**

This work proposes a method to generate handwritten text lines conditioned on given writing style and textual content. The key challenge of this task is to capture both the intra-word and inter-word style of the style sample, while maintaining the content correctness. According to the authors, most previous works focus on generating isolated words and therefore overlook the style among different words.

To capture the inter-word style, this work proposes a CNN-Transformer style encoder with two heads: a vertical head and a horizontal head. The purpose of vertical alignment is to place words in a text line on the same horizontal line, while horizontal alignment is to place words with proper spacing. To maintain content correctness, two-level discriminators are proposed to ensure the character order of the text line and word-level structure.

**Strengths:**

The method is well-motivated and clearly introduced. As far as I can see, the proposed style learning technique is interesting and different from previous methods. From experimental results, the proposed method dramatically outperforms previous methods under different evaluation metrics on English and German text-line datasets.

**Weaknesses:**

1. The reported results in Table 1 are inconsistent with the intuitive feeling. From the visualizations in Fig. 5, 12, 13, 14, text line images generated by TS-GAN are obviously better than CSA-GAN and One-DM, but TS-GAN obtain the worst quantitive results.
2. Since Unicode can encode all characters, and this paper also claims to propose a general text line generation method, it would be more convincing to conduct experiments using Chinese or Japanese. English and German only contain fewer than 100 character categories, and their structures are relatively simple; whereas Chinese and Japanese consist of thousands of characters, and their structures are complex.
3. It is better to give a experimental comparison between a CTC based content discrimnator and the proposed discrminator to support the argument in Introduction and Section 3.3.

**Questions:**

1. Why is the background color inconsistent in the style reference image and the generated images ?
2. About the line-level content discriminator, how to make sure that I_line and x_real are aligned? How to choose n? Why does Equ.(5) help to maintain the correct character order in the generated text line. It is better to give more detailed explanations.

---

### Official Review · Reviewer_TQB3 · 2024-11-01

**Soundness:** 3
**Presentation:** 3
**Contribution:** 3
**Rating:** 5
**Confidence:** 4

**Summary:**

The paper proposes a model for the generation of the images of handwritten lines, conditioned on the style sample and the desired textual content.

The model is a diffusion-based GAN, with two discriminators that analyze word and line level, acting to improve the content and the style correctness.

The style extraction model encodes specifically the information about vertical and horizontal offsets between the words by learning the information extracted from either horizontal or vertical random patches from the style source image (thus guaranteeing by construction that the information that is learned would relate to horizontal components of the style, regardless of their vertical positioning, and vice versa).

The evaluation is performed on CVL and IAM datasets, showing state-of-the-art performance in terms of CER/WER when recognizing the generated images, and HWR metric (which is somewhat similar to FID)

**Strengths:**

Originality: While there have previously been works on the generation of images of handwritten lines, using GANs and diffusion models for generation of images of handwritten words, this is (one of the) first to combine all 3.
Clarity & quality: The writing is clear, there is some amount of ablation studies to highlight the importance of the proposed components (namely the specific approach to style extraction module and the need for two different discriminators). The human preference study is also performed, showcasing that this approach is preferred 150% compared to the next one.

**Weaknesses:**

* Measuring the effect of the proposed ideas. The paper proposes the style extraction model with very set of biases, but the effect of it compared to a much simpler model that simply passes information from the style source image is not measured. Furthermore, the effect of the style model on the recognizability of the generated image seems very small as per Figure 6. Having more of such ablations would strengthen the paper.

* Generalization of the model to other data. The IAM and CVL datasets both feature fairly clear background and thin writing on top of it, making it hard to judge whether the model would generalize well for more difficult writing or backgrounds.

* Ease of reproduction. The model is fairly complex, with a custom style extractor, and reproducing the results might be not trivial. It would strengthen the paper to release the training code or the model.

**Questions:**

Results in Table 1 suggest CER of 40+% for TS-GAN and CSA-GAN, suggesting that almost half of the characters should be unrecognizable in the samples generated for these models. However, when looking at all of the results presented in Figure 5, 12, 13, and 14, there don't seem to be any errors in the results generated by these methods. Can you explain why these CER numbers are so high? This also doesn't seem to match CER numbers reported in CSA-GAN (the OCR system from which is used by the authors for the reference), which are closer to 10% (https://arxiv.org/pdf/2204.05539).

The question above is my main concern about the experimental results, happy to upgrade the rating in case of a clear answer there, and to further increase it based on the suggestions in "weaknesses" section.

---

### Official Review · Reviewer_LQQF · 2024-11-02

**Soundness:** 2
**Presentation:** 2
**Contribution:** 2
**Rating:** 5
**Confidence:** 5

**Summary:**

The paper presents  “DiffBrush”, a diffusion model devised for generating realistic handwritten text lines, it claims to address the limitations of traditional methods that focus primarily on isolated words. The authors proposed a dual-head style module that captures both vertical and horizontal style elements and a two-level content discriminator framework to ensure both style fidelity and content readability.  The paper introduces a unique “dual-head style module” for capturing vertical and horizontal writing styles independently. This module addresses alignment and spacing, crucial for generating realistic text lines that mimic human writing patterns, which are often ignored by other models focused on isolated words. Experiments were conducted on two publicly available datasets(IAM and CVL).

**Strengths:**

The authors evaluate DiffBrush using multiple quantitative metrics, such as  "Handwriting Distance (HWD)"  for style fidelity, "Character Error Rate (CER)" and  "Word Error Rate (WER)" for content accuracy, and image quality metrics (FID, IS).

**Weaknesses:**

1. Not much discussion is available  on the interpretability of the learned style space. It is not clear  how distinct are the learned vertical and horizontal style representations, and how do they vary across writers? Visualizations  of the learned style features could enhance understanding and trust in the model’s style-capturing ability.

2. More detailed explanation on how procurement of two style representations  to clearly explain how they are different from the method proposed in ONE-DM is needed.

3.DiffBrush conducted experiments  on English datasets (IAM and CVL), its performance on other languages or scripts, such as Arabic, Chinese, or Cyrillic, remains unexplored.

**Questions:**

1. It is not clear from the text how the proposed method is different from One-DM method published in ECCV 2024 in the context of blender  module (style content fusion module of One-DM).

2. Can the model generate handwritten styles that were not present in the training dataset, given only a few sample images of a writer's handwriting? Clarification on this would help in understanding the model's flexibility in handling new, unseen handwriting styles.

3. To understand the model's generalization capability, it is desirable to present results on out-of-vocabulary (OOV) text. OOV text refers to text lines that were not part of the training dataset.  It is not clear If the model can successfully generate such text, and if the modek can do so then where in the paper are the results for out-of-vocabulary text provided?

4. Figure 6 presents a table that highlights OCR results on the generated data, showing significant improvements in the last two lines. These improvements are attributed to the line-level and word-level losses. The architectural components responsible for calculating these losses are the Word-Level Discriminator and Line-Level Discriminator shown in Figure 4. However, there is not enough detailed explanation provided about these two components. The authors should offer a comprehensive description of these components, including their input, output, and intermediate tensor dimensions, for better understanding.

5. What inputs are used for the line-level and word-level content discriminators, and what do they contain during loss calculation? if it is  the final clean image, does generating it involve the full inference process with hundreds of steps, potentially adding significant overhead and increasing training time by several folds?

---

### Official Review · Reviewer_Fpfu · 2024-11-02

**Soundness:** 2
**Presentation:** 2
**Contribution:** 1
**Rating:** 3
**Confidence:** 4

**Summary:**

The paper introduces DiffBrush, a diffusion model for generating realistic handwritten text lines. The paper claims that DiffBrush improves on previous methods by capturing both vertical and horizontal writing styles and ensuring content accuracy through dual-level discriminators. Experiments show that DiffBrush generates style-consistent handwritten text lines, outperforming existing models in both visual quality and content readability.

**Strengths:**

(1) This paper introduces a diffusion model for generating handwritten text lines that adeptly captures both vertical and horizontal writing styles, using dual-level discriminators to ensure content accuracy. However, the approach encodes handwriting style in terms of intra- and inter-word spacing, which is somewhat unusual. Additionally, the encoding of styles through space with column sampling and the design of the proxy anchor loss are unclear, as noted in the weaknesses section.

**Weaknesses:**

(1) The formulation of \( L_{ver} \) and \( L_{hor} \) as proxy anchor losses is somewhat unclear. These losses appear to assume uniform spacing between words; however, this spacing is often inconsistent for a given writer. For instance, in Figure 2(a) (top row), the space between the first two words differs from that between the last two words. Furthermore, if this focuses solely on word-to-word spacing, how is character-to-character spacing addressed?

(2) The proposed method explicitly focuses on vertical and horizontal spacing to model handwriting style. However, handwriting style encompasses more than just spacing, including factors like writing speed and pressure, which are not considered in this paper. Additionally, handwriting generation could be approached as a sequence of strokes (online) rather than as static images (offline), an aspect that the paper does not address. A relevant paper can be found here: https://openreview.net/pdf?id=1ROAstc9jv.

(3) The backgrounds of the generated handwriting samples are inconsistent (see Figs. 5-8 with zoomed in), with areas behind the characters appearing slightly darker than the spaces between words. This discrepancy is not clearly explained in the text, but it may be due to the stylization process capturing background elements from the style exemplars. For realistic synthesis, however, the background should be uniform. For example, if historical handwritings from datasets like BH2M (mentioned below) were used as references, the generated handwriting backgrounds would appear unnaturally varied, which would be unacceptable.

(4) The qualitative results include only two styles, which are somewhat similar to each other. It would be interesting to consider handwritten lines from the BH2M dataset (http://dag.cvc.uab.es/the-historical-marriages-database/) as styling examples for greater diversity.

**Questions:**

(1) How do the authors envision extending the model to generate online handwriting?

---

### Official Review · Reviewer_Rzj9 · 2024-11-03

**Soundness:** 2
**Presentation:** 1
**Contribution:** 1
**Rating:** 3
**Confidence:** 4

**Summary:**

Authors propose a  tailored for handwritten text-line generation. The proposed method contains  a dual-head style module that
captures both vertical and horizontal writing styles. To make it work better, a  two-level content discriminators are introduced, aiming to  supervise textual content at both the line and word levels while preserving style imitation performance. Author conduct expensive experiments  on two widely-used handwritten datasets verify the effectiveness.

**Strengths:**

1. Authors focus on the  handwritten text generation in the wild.  The work decomposes text-line content preservation across numerous characters into global context supervision between characters and local supervision of individual character structures.

2. A lot of experiments are conducted to support the proposed method, which includes  two widely-used handwritten datasets.

3.  Authors consider more baselines, which is effective and reasonable.

4. The paper exhibits some good figures, which is clear.

**Weaknesses:**

1. I think the presentation in this paper is not good. Like ' It is non-trivial to accurately capture writing styles from text-lines with multiple words, as it involves not only intra-word style patterns like letter connections and slant but also inter-word spacing and vertical alignment' , In this paper , there are more sentences which is not readable.

2. It is hard to follow the story.  The main idea is not easy to grasp when I try to read both introduction and  the method sections.

3. Authors use more long sentences, which is easy to be wrong.  I would like to recommend authors to use shore sentence.

**Questions:**

The presentation is a big question.

---

### Official Review · Reviewer_DkjX · 2024-11-03

**Soundness:** 4
**Presentation:** 3
**Contribution:** 4
**Rating:** 8
**Confidence:** 4

**Summary:**

This paper presents DiffBrush, a novel diffusion-based approach to handwriting line generation that incorporates a dual-head style module and two-stage content discriminators to address the challenges of realistic handwriting synthesis. This model captures vertical and horizontal style features using a proxy loss method that encourages the style encoder to learn different writing patterns, ensuring accurate vertical alignment and horizontal spacing. The use of two-level content discriminators - operating at both the line and word level - enhances content monitoring by verifying global contextual coherence and local content authenticity. Extensive experimentation on 2 popular handwriting datasets using 7 different metrics demonstrates that DiffBrush outperforms existing state-of-the-art methods.

**Strengths:**

- The model extends beyond the generation of isolated words to the generation of full text lines, which is crucial for real-world applications such as synthetic data generation.
- Extensive testing is performed on two different datasets in English and German.
- The evaluation is robust, using three different sets of metrics that assess feature-based, OCR and visual quality aspects.
- Competing models are retrained to ensure a fair and direct comparison.
- The proposed method demonstrates significant performance improvements over existing alternatives.
- The study includes an ablation analysis, with both illustrative examples and quantitative assessments.

**Weaknesses:**

- Little information is provided about the OCR system used, although this is a key evaluation metric. This raises the question of whether the OCR could be specifically designed to favour the proposed generation method.
- The data sets used for the experiments are relatively simple and somewhat artificial, consisting of non-spontaneous writing with isolated words on a white background. A demonstration of the model's generalisability to more realistic, complex use cases would have strengthened the evaluation.

**Questions:**

- L315: "The proposed two-level discriminators consist of a text-level discriminator and a word-level discriminator" – Should "text-level" be corrected to "line-level"?

- L321: Can you clarify what is meant by "3D" in "3D-CNN"?

- L305: "The advantage of our discriminators is that they improve content accuracy without disrupting style learning, while CTC-based methods tend to hinder it" – Could you explain why your use of CTC in the Word-level Content Discriminator does not have this drawback?

- L334: Why is the Word-level Content Discriminator necessary?

- L372: Could you provide more details about the OCR system used?

- L377: "Resent18" – should this be "ResNet18"?

- L382: "The model is trained for 800 epochs on eight RTX 4090 GPUs using the AdamW optimizer with a learning rate of 10−4" – Can you provide an estimate of the training time?

- L384: "For the sampling ratio ρ, we perform a grid search over {0.25, 0.5, 0.75, 1.00} and ultimately set ρ to 0.25" – Since 0.25 is the lowest value tested in the grid search, how sensitive were the results to this parameter? Should you have considered values lower than 0.25?

- Figure 5: The word "destination" in your generation appears to be missing a letter and is not highlighted with a red circle. Could you comment on this?

- Notably, in Figure 5, your system seems to replicate an artifact seen in the IAM database generation, where isolated words are pasted on a white background. The background within the words is not fully white, whereas the outside background is white. Could you elaborate on why this occurs?

---

### Note · Authors · 2024-11-13

I have read and agree with the venue's withdrawal policy on behalf of myself and my co-authors.